# The Effect of a Virtual Reality-Based Intervention Program on Cognition in Older Adults with Mild Cognitive Impairment: A Randomized Control Trial

**DOI:** 10.3390/jcm9051283

**Published:** 2020-04-29

**Authors:** Ngeemasara Thapa, Hye Jin Park, Ja-Gyeong Yang, Haeun Son, Minwoo Jang, Jihyeon Lee, Seung Wan Kang, Kyung Won Park, Hyuntae Park

**Affiliations:** 1Department of Health Sciences, Graduate School, Dong-A University, Busan 49315, Korea; 2Laboratory of Smart Healthcare, Dong-A University, Busan 49315, Korea; 3Data Center for Korean EEG, College of Nursing, Seoul National University, Seoul 03080, Korea; 4Department of Neurology, Dong-A University College of Medicine, Busan 49201, Korea

**Keywords:** virtual reality, dementia, mild cognitive impairment, electroencephalogram

## Abstract

This study aimed to investigate the association between a virtual reality (VR) intervention program and cognitive, brain and physical functions in high-risk older adults. In a randomized controlled trial, we enrolled 68 individuals with mild cognitive impairment (MCI). The MCI diagnosis was based on medical evaluations through a clinical interview conducted by a dementia specialist. Cognitive assessments were performed by neuropsychologists according to standardized methods, including the Mini-Mental State Examination (MMSE) and frontal cognitive function: trail making test (TMT) A & B, and symbol digit substitute test (SDST). Resting state electroencephalogram (EEG) was measured in eyes open and eyes closed conditions for 5 minutes each, with a 19-channel wireless EEG device. The VR intervention program (3 times/week, 100 min each session) comprised four types of VR game-based content to improve the attention, memory and processing speed. Analysis of the subjects for group–time interactions revealed that the intervention group exhibited a significantly improved executive function and brain function at the resting state. Additionally, gait speed and mobility were also significantly improved between and after the follow-up. The VR-based training program improved cognitive and physical function in patients with MCI relative to controls. Encouraging patients to perform VR and game-based training may be beneficial to prevent cognitive decline.

## 1. Introduction

Dementia is a syndrome characterized by global cognitive impairment [1] and an estimated 50 million people worldwide have dementia [2]. It is most prevalent in individuals aged > 65 years and is considered as the greatest health challenge in the 21st century [3]. The pharmacological treatments have not yet led to an important breakthrough in the treatment of dementia, resulting in gravitation toward non-pharmacological approaches to alter the progressive course of the disease [4]. Mild cognitive impairment (MCI) is the prodromal stage of dementia [5], with around 46% progressing to dementia within 3 years [6]. Some MCI patients are stable or return to a normal state over time [7]. Thus, MCI serves as an ideal stage for preventive interventions.

The most promising non-pharmacological interventions for delaying the progression of MCI to dementia are exercise, cognitive training [8] and multicomponent intervention [9], and recently, virtual reality (VR) has also been explored for treatment and prevention of dementia [10]. A randomized control trial in normal older adults comparing physical exercise, cognitive exercise and VR exercise demonstrated that VR exercise showed significant improvements in cognitive as well as physical function, and VR exercise was more favored than physical exercise by the elderly [11]. In older adults with MCI, VR intervention has been reported to improve memory function [12]. VR is a computer-simulated environment, closely resembling real-life situations and scenarios, which provides the user with the sensation of physically “being there” [13,14]. VR can be divided into non- immersion, semi-immersion, and full immersion, based on the levels of immersion. Immersive VR provides enhanced ecological validity and the possibility to personalize the activity and environment to the needs of individuals, making VR-based training more engaging [15]. The higher level of immersion corresponds to a more realistic VR environment to the user [16]. The immersive VR has been used mostly for neurophysiological assessment, cognitive rehabilitation and the effect of VR on different cognitive domains such as executive function, attention, memory and spatial orientation is being investigated [17]. 

Immersive VR-based cognitive training has recently attracted attention in the research filed of MCI and dementia [18]. A recent meta-analytic study has also reported the positive effect of semi-immersive VR on cognition and physical function in individuals with MCI and dementia [19]. A fully immersive VR-based cognitive intervention study on older adults with and without dementia reported significantly higher cognitive progress in those without dementia compared to those with dementia [20]. A feasibility study using fully immersive VR cognitive training in MCI and dementia patients showed a high feeling of safety, satisfaction, reduced anxiety fatigue and low discomfort compared to pen-paper training [15]. This advantage of VR over pen-paper cognitive training can be beneficial to ensure adherence to cognitive training and when performed with caution, it has the potential to be an efficient intervention for dementia patients [15,21]. 

Nevertheless, immersive VR is increasingly used in health-related fields and interventions and has the potential to be a powerful tool in delaying the onset of degenerative brain and mental diseases. However, the evidence regarding the effectiveness of immersive VR in MCI and dementia is limited [22], especially the fully immersive type of VR. Several studies still rely mostly on non- and semi-immersive VR [23] and the advantages of VR with full immersion and interaction on prevention of dementia have not yet been explored to their full potential. Therefore, we aimed to investigate the effect of the fully immersive virtual reality intervention on cognitive, brain and physical function in older adults with MCI. In addition to the neurocognitive tests, we also used electroencephalography (EEG) to assess the effect of VR intervention on cognition in our study. EEG is the least invasive measure of brain electrical activity. EEG power measures are associated with memory, executive function and global cognition in patients with MCI [24]. 

## 2. Materials and Methods

### 2.1. Subjects

The study was announced through three regional health care centers in Busan metropolitan city, South Korea. A total of 234 participants applied and all of them underwent a screening process. The inclusion criteria were: (i) >55 and <85 years and (ii) individuals diagnosed with MCI based on medical evaluations consisting of neurological examinations and detailed neuropsychological assessments conducted by a dementia specialist. MCI was defined by the Consortium to Establish a Registry for Alzheimer’s Disease Assessment Packet (CERAD) with the cut-off score developed by Chandler et al. [25]. Following some exclusion criteria, 100 out of 234 subjects were recruited to be part of this study. The exclusion criteria are explained in Figure 1. Thirty-two subjects out of 100 did not participate in the study due to personal reasons, hence our study included a total of 68 subjects who were then randomly allocated to either the control (n = 34) or VR-intervention (n = 34) group. To avoid selection bias, the random allocation was assigned via a computer-generated, fixed block randomization procedure to either the intervention or control group, generating blocks stratified by age, gender, education level, and participating center. The participants’ mean age was 72.5 ± 5.32 years (mean ± standard deviation (SD)).

The study procedure was approved by Dong-A University Institutional Review Board on 24 October 2019 (IRB No. 2-1040709-AB-N-01-201909-HR-036-04) and all participants provided signed informed consent at the beginning of the study. This study is registered in the University Hospital Medical Information Network (UMIN) Clinical Trials Registry (Registration No. UMIN000040107).

### 2.2. Intervention

The intervention group performed a total of 24 sessions of VR-based cognitive training for eight weeks. Three sessions were held per week and each VR training session lasted for 100 minutes, which also included instruction regarding VR training and eye stretching exercises in between VR training, as described in Figure 2. On the other hand, the control group participated in an educational program on general health care once a week during the study intervention period (8 sessions). Each session was 30 to 50 minutes. The program was led by health professionals, an exercise specialist, a physical therapist and a nutritionist, and subjects were given information such as nutrition regarding proper diet and foods, and exercise tips to prevent geriatric diseases such as frailty, sarcopenia and dementia. In addition to VR training, the intervention group was also provided with an educational program following the same protocol as the control group.

The VR training consisted of 4 series of games. All the VR training game contents were developed by SY Innotech Inc., Busan, South Korea. The VR training was performed with an Oculus VR headset (Oculus quest headset) and two wireless hand controllers, one for each hand. The virtual reality program consists of four types (Figure 2): Juice making: This game requires the subject to pick a recipe for making a given juice in a virtual reality space, memorize the listed fruits, select the memorized material from the shelf and toss it into a container. The container is vigorously shaken until juice is made and then poured into the cup. Crow Shooting: The shooting game is set up on the beachside where the subjects are asked to shoot flying black birds. The right controller turns into a gun and the left controller into a shield. Fireworks (find the fireworks number): Three numbers are shown on the screen when the game starts and fireworks burst in a random order from the numbers. When the firework display is over, the subjects need to click the numbers in the order of which the firework exploded. Love house (d: memory object at the house): In this particular game, subjects are placed in a virtually simulated house where they were given 30 seconds to memorize the location of objects in the shelf and around the living area. The items are then misplaced, and the subjects are required to reorganize the objects in the correct location with the help of a VR hand controller.

### 2.3. Cognitive Function

Cognitive assessments were performed by neuropsychologists according to standardized methods. Global cognitive function was assessed with the Korean version of the Mini-Mental State Examination-Dementia screening test (MMSE-DS) [26]. From the National Center for Geriatrics and Gerontology Functional Assessment Tool (NCGG-FAT), the trail making test (TMT) A & B, and symbol digit substitution test (SDST) in the form of an electronic tablet were used to assess cognitive function [27]. In TMT A, the subjects were instructed to touch the target number in consecutive order of 1–15 shown in the tablet as quickly as possible. In TMT B, the subjects were instructed to touch numbers and letters alternatively in consecutive order. In SDST, 9 pairs of numbers and symbols were shown in the top half of the tablet display, and the subjects were asked to match the symbols to their corresponding numbers displayed in the bottom half of the display. TMT and SDST were both timed, and shorter time taken to complete the test indicated better cognitive function.

### 2.4. EEG Recording and Data Acquisition

The EEG was recorded in a dimly lit and quiet room with a Cognionics Quick-20 (Cognionics Inc., San Diego, CA, USA) dry EEG headset. The EEG headset had a built-in design based on the international 10–20 system (Fp1, Fp2, F7, F3, Fz, F4, F8, T3, C3, Cz, C4, T4, T5, P3, Pz, P4, T6, O1 and O2) positioned according to the international 10–20 system [28]. The resting state EEG data of 5 minutes was recorded with the eyes closed condition. The subjects were instructed to keep their eyes closed but stay awake during EEG recordings. The EEG was recorded at a sampling rate of 500 Hz and filtered through a band-pass of 0.53–120 Hz. The electrodes impedance was kept under 500 kΩ. The artifact removal was conducted with independent component analysis (ICA) individually for each channel. The EEG brain power ratio-theta/beta ratio (TBR), theta/alpha ratio (TAR) and delta/alpha ratio (DAR) between baseline and follow-up data (within the VR intervention and control groups) were analyzed in iSyncBrain software, v.2.1, 2018 (iMediSync, Inc., Seoul, South Korea) and the difference was observed with paired t-test. The EEG band power was shown in a topographic map (Topo map) plotted with the built-in sLoreta function in iSyncBrain software v.2.1. The differences were observed with paired t-test. 

### 2.5. Physical Function

Physical function was assessed by gait speed test, mobility test and handgrip strength. The gait speed test was 7 m long, which included a 1.5 m acceleration distance, a 4 m “preferred walking speed” followed by a 1.5 m deceleration distance. Only the 4 m walk was timed. The 8-feet Up and Go test of 2.44 m was used to assess mobility. Handgrip strength (HGS) of the non-dominant hand was measured with a digital hand dynamometer (TKK 5101 Grip-D Takei, Tokyo, Japan). The HGS was measured twice and the mean value was used for statistical analysis. During the test, participants were instructed to maintain their shoulders slightly apart from their body, and hold the dynamometer pointing to the ground.

Socio-demographic variables such as education, age, medication intake and smoking were acquired through interviews by trained researchers.

### 2.6. Statistical Analysis

All comparisons were two-sided, with an alpha level of 0.05. All statistical analyses were analyzed using the IBM SPSS Statistics, version 25.0, 2017 software package for Windows (SPSS Inc., Chicago, IL, USA). The Shapiro–Wilk test was used to determine the normality of the data distribution. The independent t-test or the chi-square test was used to assess differences in the baseline (beginning of the intervention) variables. We used the intention-to-treat approach, and between-group comparisons of continuous variables were conducted using a repeated measures analysis of variance (ANOVA) model after adjusting for the potential covariates (age, sex and years of education) for primary and secondary endpoint outcome. Time was treated as a categorical variable, and the models included group, time and group-by-time interaction as fixed effects. The conclusions about the effectiveness of the VR intervention were based on between-group comparisons of change in global and prefrontal cognitive function (working memory, processing and executive function) from baseline to 8 weeks after, as assessed with the MMSE, TMT A & B, SDST and other physical function determined by the time-by-group interaction of the model.

## 3. Result

In Table 1, the demographics of the study population along with baseline physical functions and global cognitive functions are described. There are no significant differences in parameters between the intervention and control groups at the baseline.

In Table 2, TMT B time decreased significantly in the intervention group compared to the control group (*p* = 0.03). Similarly, small, but not significant, positive changes were observed in MMSE and SDST. The physical function such as gait speed (*p* =0.02) and 8-feet Up and Go were significantly improved (*p* = 0.03) in the intervention group.

### 3.1. EEG

#### 3.1.1. Band Power

In the elderly, increased Theta wave (>3.5 to <8 Hz) is associated with the risk of developing cognitive impairment [29,30]. The theta has been observed to be significantly decreased around the parietal (*p* = 0.013) and temporal (*p* = 0.036) regions at follow up (G2) compared to baseline (G1) in the VR intervention group (Figure 3).

#### 3.1.2. Power Ratio

The higher TBR is related to mind wandering, which is considered to be associated with reduced attention [31]. In the VR intervention group, the Theta/Beta ratio (TBR) is decreased in the temporal (*p* = 0.035) and parietal (*p* = 0.027) regions at follow up (Figure 4). In the control group, all power ratios did not show any changes.

## 4. Discussion

Our study examined the effect of a VR game intervention on cognitive and brain activity in older adults with MCI. Our findings show that the VR game intervention is an effective way to improve cognitive and frontal brain function in MCI patients. The improvements in the 8-feet Up and Go test and gait speed were also observed in the intervention group.

Over the past decade, technology-based cognitive interventions have gained immense interest worldwide. Our study results show positive effects for the VR intervention on key outcome variables, such as cognitive and physical function. The VR intervention effect in the cognition category is consistent with the result of a systematic review by Coyle et al. [32]. Coyle and colleagues showed that the VR intervention moderately improved the cognitive function of participants with cognitive impairment. A recent VR study has reported improvements in executive functions [33] after the VR intervention, similar to our study results. TMT is often used as a measure of executive function [34]. TMT A measures psychomotor speed and visual scanning, while TMT B reflects working memory [35,36,37]. However, our results showed significant positive improvements in TMT B but not in TMT A. The contents of our VR training consisted of games such as love home, juice making and fireworks, which required subjects to memorize objects, recipes and numbers respectively, highly involving use of working memory along with other cognitive functions, such as attention and processing speed. This could explain the significant change in TMT B score. In addition, we also measured brain electrical activity during the resting state of mind. The theta power was observed to be decreased in the VR intervention at follow up. Increased theta, in the elderly, is associated with the risk of developing cognitive impairment [29,30]. Our study also a showed significant decrease in TBR in the temporal and parietal regions of the brain. The higher TBR was observed to be related to mind wandering, which is considered to be associated with reduced attention [31]. However, detailed studies over a longer period on each EEG rhythm are required to draw its strong relationship with VR training.

Studies have also shown that VR intervention is effective to improve physical function and walking speed of community-dwelling patients who had a stroke [38,39,40]. Our study results further expand this data as significant improvements in physical function, especially gait speed and the 8-feet Up and Go test, in MCI patients were observed after 8 weeks of VR training. Several studies have found physical functions such as handgrip strength [41,42,43] and gait speed [44] to be associated with cognitive decline in older adults. Gait speed has been reported to rely on motor function and cognitive processes, which includes executive function and attention [45], which may explain the relationship between gait speed and cognition. Furthermore, improved brain function may be due to health promoting behaviors, such as physical activity and nutrition, and health literacy [46]. However, further studies are needed in order to evaluate the bidirectional relationship between the improvement of cognition and physical function.

VR provides an artificial interactive environment closely representing reality. Older adults with dementia can experience various sensory stimulation in a comfortable and safe virtually simulated environment, which could lead to a boost in functional learning and transfer of learned functions [47]. Unlike computerized computer training, VR is immersive and closely mimics reality, intensifying ecological legitimacy, offering greater potential for transfer to activities of daily life (ADL) [48]. We performed fully immersive VR training in order to examine the effectiveness of using VR intervention on cognitive, brain and physical function in MCI patients. The VR contents in our study used real life locations. Fully immersive VR corresponds more to feeling like the individual is experiencing real-life scenarios [23]. A functional magnetic resonance imaging study showed that the virtual reality-generated environment activates the associated brain areas, similar to the real environment [49].

Our study yielded a positive effect of fully immersive VR training on cognitive as well as physical function in older adults with MCI. The subjects in the VR intervention group strongly adhered to the training and the dropout number was low (n = 1). One of the limitations of our study was that there was no follow up in the middle of the intervention period or some period after the study was over, due to which the short- and long-term effect of the VR intervention on cognition and physical function were not observed. Secondly, the VR intervention program group had more sessions (n = 24) compared to the control group (n = 8) and hence, the intervention group had more interaction with each other and their supervising health professionals. The increased social interaction has been associated with preventing cognitive decline [50,51]. This could have somewhat contributed to positive results in the intervention group. The participants of our study were predominantly women (n = 52), compared to men (n = 16), and future studies regarding gender differences on the effect of the VR intervention are necessary.

## 5. Conclusions

In summary, our results show that VR-based cognitive training has a positive effect on cognition in MCI patients. Although the global cognitive function did not change significantly, we observed significant improvement in executive function, and some physical functions such as gait speed and the 8-feet Up and Go test. Moreover, the EEG test showed a positive change in brain activity related to attention after the intervention period. Nevertheless, further work is needed in this area to confirm the long-term effectiveness and feasibility.

## Figures and Tables

**Figure 1 jcm-09-01283-f001:**
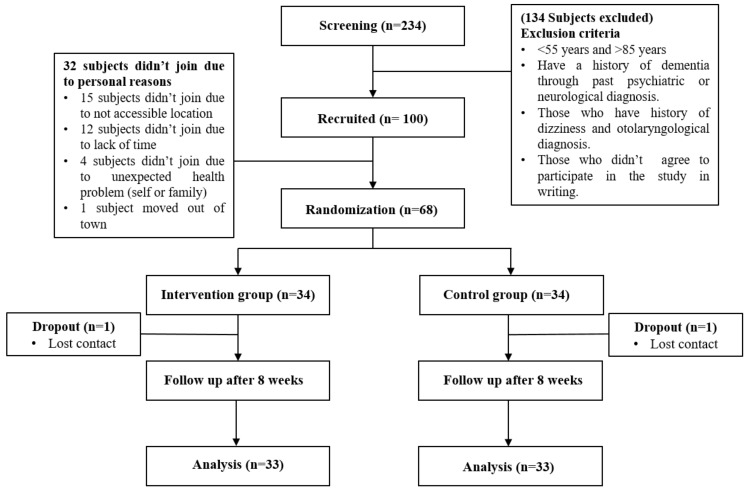
Flowchart of study design.

**Figure 2 jcm-09-01283-f002:**
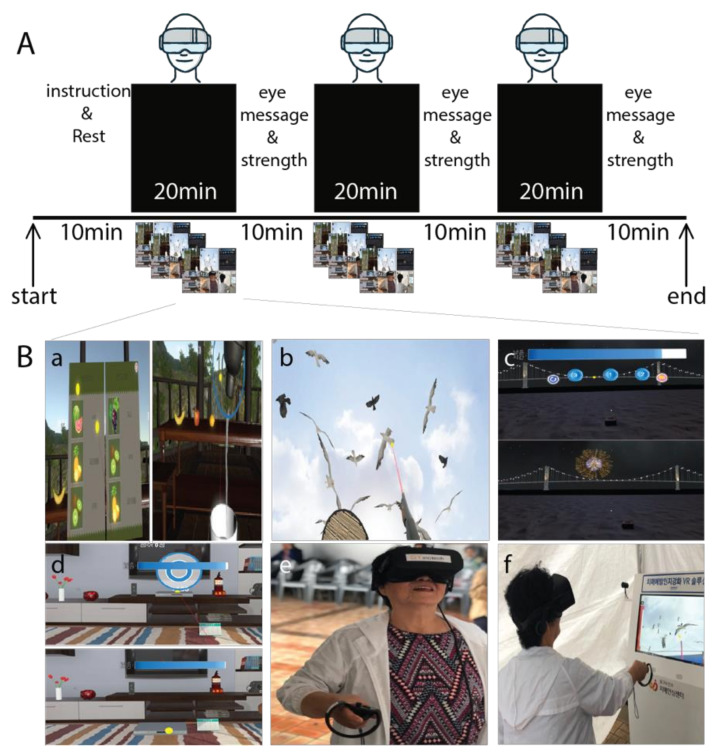
(**A**) This figure illustrates the design for the virtual reality (VR) training design and game contents. The total training duration was 100 minutes (three 20 min VR training sessions, and three 10 min eye massage and stretching sessions) held three times a week for 8 weeks. (**B**) The contents of the VR training games: (a) juice making, (b) crow shooting, (c) find the fireworks number, (d) memory object at the house, (e) and (f) example of subject.

**Figure 3 jcm-09-01283-f003:**
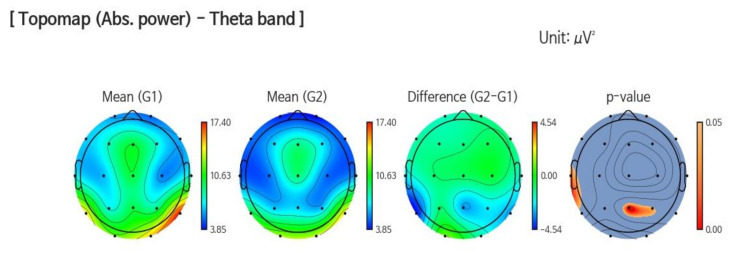
This figure shows a significant decrease in Theta power in the parietal (*p* = 0.013) and temporal areas (*p* = 0.036) at follow up (G2) compared to baseline (G1) in the VR intervention group.

**Figure 4 jcm-09-01283-f004:**
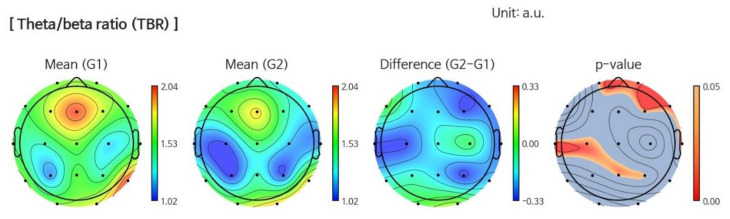
This figure shows a significant decrease in the Theta/Beta power ratio (*p* = 0.027) at follow up (G2) compared to baseline (G1) in the VR intervention group.

**Table 1 jcm-09-01283-t001:** Selected anthropometric, cognitive and physical function characteristics of the subjects at the baseline.

Variables	VR Intervention	Control
n (male)	34 (6)	34 (10)
Age (years)	72.6 ± 5.4	72.7 ± 5.6
Education (years)	9.3 ± 4.0	8.4 ± 3.5
Height (m)	1.58 ± 0.08	1.58 ± 0.08
No. of medication intake (n)	2.3 ± 1.4	2.12 ± 1.4
Weight (kg)	60.7 ± 9.8	61.3 ± 9.1
BMI (kg/m^2^)	24.3 ± 3.0	24.5 ± 2.7
SBP (mg/hg)	129.6 ± 15.8	129.6 ± 17.9
DBP (mg/hg)	74.8 ± 11.3	69.5 ± 11.8
Grip strength (kg)	22.2 ± 6.3	23.4 ± 5.7
Gait speed (s)	1.15 ± 0.33	1.18 ± 0.21
8-feet Up and Go (s)	6.27 ± 1.48	7.04 ± 2.02
MMSE (score)	26.0 ± 1.8	26.3 ± 3.3
TMT A (s)	26.3 ± 7.3	27.9 ± 9.2
TMT B (s)	56.6 ± 25.0	58.5 ± 28.1
SDST (score)	33.4 ± 9.0	32.4 ± 8.2

BMI: Body mass index, SPB: Systolic blood pressure, DBP: Diastolic blood pressure, MMSE: Mini mental state examination, TMT A: Trail making test A, TMT B: Trail making test B, SDST: Symbol digit substitution test. The values are expressed in mean and standard deviation (mean ± SD). All variables have no significant differences measured by independent t- test or chi-square test.

**Table 2 jcm-09-01283-t002:** The comparison of physical function and global cognitive function between baseline and post intervention in the VR intervention and the control groups.

Variables	VR Intervention			Control			Group x Time Interaction	
	Baseline	Follow Up	*p*-Value ^a^	Baseline	Follow Up	*p*-Value ^a^	*p*-Value ^b^	Effect Size
Grip strength (kg)	22.2 ± 6.3	24.4 ± 5.3	0.03	23.4 ± 5.7	23.9 ± 5.7	n.s.	n.s.	-
Gait speed (m/s)	1.15 ± 0.33	1.19 ± 0.37	0.04	1.18 ± 0.21	1.12 ± 0.26 *	0.01	0.02	0.143
8-feet Up and go (s)	6.77 ± 1.48	6.32 ± 1.92	0.02	7.04 ± 2.02	7.06 ± 1.87 *	n.s.	0.03	0.107
MMSE (score)	26.0 ± 1.8	26.9 ± 2.0	n.s.	26.3 ± 3.3	26.4 ± 2.7	n.s.	n.s.	*-*
TMT A (s)	26.3 ±7.3	24.2 ± 5.3	0.04	27.9 ± 9.2	27.8 ± 8.1	n.s.	n.s.	*-*
TMT B (s)	56.6 ± 25.0	51.3 ± 24.8	0.03	58.5 ± 28.1	63.2 ± 25.1 *	0.01	0.03	0.208
SDST (score)	33.4 ± 9.0	39.6 ± 9.5	0.02	32.4 ± 8.2	21.8 ± 8.2	< 0.01	0.03	0.264

The values are expressed in mean and standard deviation (mean ± SD). ^a^ Paired t-test between baseline and follow-up assessment. ^b^ Repeated-measures analysis of variance (ANOVA) testing interaction of intervention (VR intervention versus control) by time (baseline versus follow-up) for each outcome, adjusted for age, gender and years of education. Effect size is partial eta squared for group by time. ******* Represents a significant difference between the intervention and control group. n.s. = not significant, MMSE: Mini mental state examination, TMT A: Trail making test A, TMT B: Trail making test B, SDST: Symbol digit substitution test.

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
