# Peer review of "The Effect of a Virtual Reality-Based Intervention Program on Cognition in Older Adults with Mild Cognitive Impairment: A Randomized Control Trial"

_jcm, 2020, doi:10.3390/jcm9051283_

Round 1
Reviewer 1 Report
The article reports a study on the use of virtual reality intervention on Mild Cognitive Impairment (MCI) patients to enhance cognitive functions. An intervention group is compared with a control group that receives no intervention. Significant differences are found and it is concluded that VR-based interventions are helpful for therapy in MCI patients.
The article is written in a classically structured and focused manner and as such is scientifically sound. However, there are some aspects that should be described IMHO in more detail. These are mentioned in the following:
- What is the condition of the control group? Did they receive no treatment? IMHO it is not fair to compare treatment with no treatment, find some statistical differences and conclude that VR shows effects on MCI patients. Fair would be to compare VR with another treatment, e.g. paper-based treatments (e.g. solving printed riddles). Please elaborate on why you have chosen the VR treatment and provide a literature review on other possible treatments for MCI. One key question is if VR is the only possible treatment and if the technology of VR is required (or if there would be effective treatments with lower technological hurdles)
- What were the criteria to assign the subjects to the treatment group or to the control group? Randomly?
- Table 1: Both groups contain only 20 % male subjects. Why this unbalanced selection? What is the impact on the results? They are valid only for female subjects? What are the results if the male subjects are eliminated?
- Table 2: Did you expect on grip strength through the treatment? It would be helpful to explain why you chose these measures. Further, it would be beneficial for the understanding to discuss if the results are consistent (Just to exclude a random result, where some of many dependent variables show significant changes, but others do not).
- L59: “A total of 234 participants applied and all of them underwent screening process.” How was the study announced? Did the elderly apply themselves or “were they applied” by someone else? Why are there exclusion criteria, such as “do not agree in the study”, after they have applied to the study?
- L63: “Twenty five subjects dropped out in the beginning of the study.” When? (What is “in the beginning”?) What were the reasons?
- L64: 75 subjects, Figure 1: 68 subjects: Why is there a difference?
- L78: Oculus VR headset? What kind of VR glasses? Oculus provides different types of VR headsets.
- L119: “international 10–20 system“ is there a reference?
- Figure 4: Topomaps have been created using which software?
- L218 – L252 does not fit into the conclusion sections, it belongs more to Introduction / Literature review
- Probably for the literature review: The benefits of emotion regulation interventions in virtual reality for the improvement of wellbeing in adults and older adults: a systematic review (https://www.mdpi.com/2077-0383/9/2/500)
Formal issues
- Title lower/upper letter mix: should be completely in TitleCase:
The Effect of Virtual Reality (VR)-based Intervention Program and cognition in older adults with Mild Cognitive impairment (MCI): A randomized control trial -> The Effect of Virtual Reality (VR)-based Intervention Program and Cognition in Older Adults with Mild Cognitive Impairment (MCI): A Randomized Control Trial
- Title: The effect of VR on Cognition … instead of The effect of VR and cognition?
- Author: Hyuntae park -> Hyuntae Park ?
- L74: “The intervention group performed of VR- based cognitive training for eight weeks.” -> “The intervention group performed an VR- based cognitive training for eight weeks.”
- L82: “The container is until juice is made and poured into the cup.” What does this mean? I do not understand.
- L111: “TMT and DSST were both timed test where shorter time taken to complete the test indicated better cognitive function” Please rephrase.
- L172 „3.2 EEG\“ -> „3.2 EEG“
- Please pay attention to the spelling (the following mistakes stem only from L214 to L231, there are many other easy-to-avoid errors)
- L214: “MCI patient” -> “MCI patients”, further: “In previous study“ -> “In a previous study” or “In previous studies”?
- L224: “might be due the to contents” -> “might be due to the contents”
- L225 “fireworks, had high demand for the use of cognititve funcitons such asworking memory and attention” -> “fireworks, had high demand for the use of cognititve functions such as working memory and attention”
- L226 “and processing speed.A recent VR study has aslo reported improvements in executive fucntions [14]” -> “and processing speed. A recent VR study has also reported improvements in executive functions [14]”
- L226 “In this study we measued brian wave activites during resting rate of mind.” -> “In this study we measured brain wave activities during the resting state of mind.”
- L231: “Our finding show“ -> “Our findings show” or “Our finding shows”?
- Please mind double spaces (e.g. in L240)
- Careful proofreading, preferable by a native speaker, is required.
Author Response
Reviewer #1
Comment 1: What is the condition of the control group? Did they receive no treatment? IMHO it is not fair to compare treatment with no treatment, find some statistical differences and conclude that VR shows effects on MCI patients. Fair would be to compare VR with another treatment, e.g. paper-based treatments (e.g. solving printed riddles).
Response 1: Thank-you for the thorough review of our paper. We are appreciate your insightful comments. We performed the educational treatment program in the study duration as follows and we added the information of the program in the manuscript;
All the subjects included in our study had Mild Cognitive Impairment (MCI), which is one of the inclusion criteria of our study. To reduce the dropout rates in the control group, the control group were given educational treatment program (8 session, 30-50minutes by healthcare specialist) held at health care center. To avoid the bias, the intervention group also took part in same educational program protocol as control group in addition to VR intervention in the study duration.
We have added the program performed by control group as follows (Currently line 136-141): “The control group and VR training group participated in educational program on general health care once a week during the study intervention period (8 session). Each class was 30 minutes to 50 minutes. The program was led by health professionals, exercise specialist, physical therapist and nutritionist, and subjects were given the information such as, nutrition regarding proper diet and foods, exercise tips to prevent geriatric diseases such as frailty, sarcopenia and dementia.”
Comment 2: Please elaborate on why you have chosen the VR treatment and provide a literature review on other possible treatments for MCI. One key question is if VR is the only possible treatment and if the technology of VR is required (or if there would be effective treatments with lower technological hurdles)
Response 2: Thank you for reviewing our paper. We greatly appreciate you comment.
We added background on other possible treatments for MCI and we have further explained why we have chosen VR treatment in the introduction section. We have made modification throughout the introduction section keeping the reviewer’s question and suggestion in mind as follows (Currently line 44-76):
“The most promising non-pharmacological interventions for delaying progression MCI to dementia are exercise, cognitive training [8] and multicomponent intervention [9] and recently virtual reality (VR) have also been explored for treatment and prevention of dementia [10]. A randomized control trail in normal older adults comparing physical exercise, cognitive exercise, and VR exercise demonstrated that VR exercise showed significant improvements in cognitive as well as physical function and VR exercise was most favored than physical exercise by elderly [11]. In older adults with MCI, VR intervention has been reported to improve memory function [12]. Virtual reality (VR) is a computer stimulated environment, closely resembling real-life situations and scenarios, which provides user with sensation of physically “being there” [13,14]. VR can be divided into non- immersion, semi-immersion, and full-immersion based on levels of immersion. Immersive VR provides enhanced ecological validity and possibility to personalize the activity and environment to the need of individual making VR-based training more engaging [15].The higher level of immersion corresponds to more realistic VR environment to the user [16]. The immersive VR have been used mostly for neurophysiological assessment, cognitive rehabilitation and effect of VR on different cognitive domains such as executive function, attention, memory and spatial orientation is being investigated [17].
Immersive VR based cognitive training has recently attracted attention in research filed of MCI and dementia [18]. A recent Meta analytical study have also reported positively effect of semi-immersive VR on cognition and physical function in individuals with MCI and dementia [19]. Fully immersive VR based cognitive intervention study on older adults with and without dementia reported significantly higher cognitive progress in those without dementia compared to those with dementia [20]. A feasibility study using fully immersive VR cognitive training in MCI and dementia patients showed high feeling of security, satisfaction and reduced anxiety, fatigue, and low discomfort compared to pen-paper training [15]. This advantage of VR over pen-paper cognitive training can be beneficial to ensure adherence to cognitive training and when performed with caution, it has potential to be an efficient intervention for dementia patient [15,21].”
Comment 3: What were the criteria to assign the subjects to the treatment group or to the control group? Randomly?
Response 3: Thank you for the time and effort you put in our paper. We found your comment very helpful to make our paper more understandable. We have added more information regarding group assignment. We hope that it provides clear understanding of group assignment in our study.
The modified sentence now reads as (Currently line 115-120): “Thirty two subjects out of 100 didn’t participate in the study due to personal reasons, hence our study included total 68 subjects who were then randomly allocated to either Control (n=34) or VR- intervention (n=34) group. To avoid selection bias, the random allocation was centralized and internet-based, generating blocks stratified by age, gender, education level and participating center. The participants’ mean age was 72.5±5.32 years (mean ± S.D.).”
Comment 4: Table 1: Both groups contain only 20 % male subjects. Why this unbalanced selection? What is the impact on the results? They are valid only for female subjects? What are the results if the male subjects are eliminated?
Response 4: Thank you. You have raised an important point here. There was a few male participation in recruiting through recruitment in this study. We have added this as one of the limitations of our research, as reviewers pointed out.
Since our study consisted both male and female subjects, statistical analysis was performed after adjusting gender and age. It was considered to perform the analysis excluding male subjects as the reviewer's suggestion, however intention-to-treat (ITT) analysis was performed to preserve the integrity of randomization and if male subjects are excluded, the randomization of our study would be broken in the statistical fit. Hence we didn’t perform separate analysis eliminating male subjects.
I think the issue of gender difference in virtual reality-based mediation is being debated. In particular there is a lack of study to find the VR intervention and gender difference in older population. A previous study reported that there is no gender difference and both men and women devoted approximately the same amount of attention to the environment, regardless of the experimental condition (virtual vs. imaginative) [1], but another reported the presence of gender difference [2].
Nevertheless, as you suggested, a small number of males may be misinterpreted as a gender difference. Hence in the future, a follow-up study involving balanced number of male and female subjects, on whether VR effects will have a gender difference is necessary.
We have added this point to the limitation on line 539-540 as follows: “The participants of our study were predominantly women (n=52) compared to men (n=16) and future studies regarding gender difference on effect of VR intervention is necessary.”
References
- Felnhofer, Anna, et al. "Is virtual reality made for men only? Exploring gender differences in the sense of presence." Proceedings of the International Society on presence research (2012): 103-112.
- Sakthivel, Madhumathi, P. E. Patterson, and C. Cruz-Neira. "Gender differences in navigating virtual worlds." Biomedical sciences instrumentation 35 (1999): 353-359.
Comment 5: Table 2: Did you expect on grip strength through the treatment? It would be helpful to explain why you chose these measures. Further, it would be beneficial for the understanding to discuss if the results are consistent (Just to exclude a random result, where some of many dependent variables show significant changes, but others do not).
Response 5: We are grateful to you for reviewing our utilizing your precious time to review our paper to provide us insightful comments.
The reason we used grip strength as one of the variables in our study was: (i) the grip strength has been associated with cognitive decline by previous studies [1, 2, 3], and (ii) Another reason was that our game contents required use of hands more for activities like pointing, shaking and grabbing controller firmly. However, we didn’t find significant result for the grip strength in our study so we did not further discuss about it.
We have added this information in the manuscript as follows (Currently line 390-395): “Several studies have found physical functions such as hand grip strength [39-41] and gait speed [42] to be associated with cognitive decline in older adults. Gait speed has been reported to rely on motor function and cognitive processes which includes executive function and attention [43] which could explain the relationship between gait speed and cognition. Furthermore improved brain function may be accounted by health promoting behaviours such as physical activity and nutrition, and health literacy [44].”
References
- Fritz, Nora E., Caitlin J. McCarthy, and Diane E. Adamo. "Handgrip strength as a means of monitoring progression of cognitive decline–a scoping review." Ageing research reviews 35 (2017): 112-123.
- Sternäng, Ola, Chandra A. Reynolds, Deborah Finkel, Marie Ernsth-Bravell, Nancy L. Pedersen, and Anna K. Dahl Aslan. "Grip strength and cognitive abilities: associations in old age." Journals of Gerontology Series B: Psychological Sciences and Social Sciences 71, no. 5 (2016): 841-848.
- Vancampfort, Davy, Brendon Stubbs, Joseph Firth, Lee Smith, Nathalie Swinnen, and Ai Koyanagi. "Associations between handgrip strength and mild cognitive impairment in middle‐aged and older adults in six low‐and middle‐income countries." International journal of geriatric psychiatry 34, no. 4 (2019): 609-616.
Comment 6: L59: “A total of 234 participants applied and all of them underwent screening process.” How was the study announced? Did the elderly apply themselves or “were they applied” by someone else? Why are there exclusion criteria, such as “do not agree in the study”, after they have applied to the study?
Response 6: Thank you for raising this question in the comments. This comment made us realize how vague our explanations were.
For this study, consultation was conducted with, a public health center, and a dementia care center under public health center in three districts in Busan. Elderly people in communities participate in community health check-ups. The public health center with consent of the dementia care center, obtained the information of the subject such as the address, contact number, and contacted them individually. In addition, in parallel with individual liaisons, the entire recruitment was also carried out through the advertising leaflets distributed in the communities.
We have added some information in the method sections as well. It reads as (Currently line 109-110): “ The study was announced through three regional health care center in Busan metropolitan city, South Korea.”
The “did not agree to participate” included participants who agreed verbally but did not sign the consent. We have modified this sentence for better interpretation in the flowchart (Figure1) as follows:
“Those who did not agree to participate in the study in writing.”
Comment 7: L63: “Twenty five subjects dropped out in the beginning of the study.” When? (What is “in the beginning”?) What were the reasons?
Response 7: Thank you for the comment. We have received comments on this from another reviewer too.
The dropout number is actually thirty two. We made an error in the manuscript. We have corrected this error and the new sentence now reads as follows (Currently line 116-122): “Thirty two subjects out of 100 didn’t participate in the study due to personal reasons, hence our study included total 68 subjects who were then randomly allocated to either Control (n=34) or VR- intervention (n=34) group. To avoid selection bias, the random allocation was centralized and internet-based, generating blocks stratified by age, gender, education level and participating center. The participants’ mean age was 72.5±5.32 years (mean ± S.D.).”
We have added the reasons not participating in the flowchart of figure 1 in bullet point as follows in the manuscript:
- 15 subjects didn’t join due to not accessible location
- 12 subjects didn’t join due to lack of time
- 4 subjects didn’t join due to unexpected health problem (self or family)
- 1 subject moved out of town
Comment 8: L64: 75 subjects, Figure 1: 68 subjects: Why is there a difference?
Response 8: Thank you for pointing this out. It helped us recognize the mistake we made while writing manuscript.
We have edited the correct subject number in the manuscript. The new sentence now reads as (Currently line 116-122): “Thirty two subjects out of 100 didn’t participate in the study due to personal reasons, hence our study included total 68 subjects who were then randomly allocated to either Control (n=34) or VR- intervention (n=34) group. The randomization was performed after adjusting variables such as age, sex, education level, and living status of participants.”
Comment 9: L78: Oculus VR headset? What kind of VR glasses? Oculus provides different types of VR headsets.
Response 9: We are thankful to you for this comment. We apologize for not specifying the device we used in the study.
We used Oculus Quest VR headset. We have also added this information in the manuscript as follows (Currently line 144-145): “The VR training was performed with Oculus VR headset (Oculus quest headset) and two wireless hand controller, one for each hand.”
Comment 10: L119: “international 10–20 system” is there a reference?
Response 10: Thank you for mentioning this point.
We have added appropriate reference in the manuscript.
(Currently line 191-194): “The EEG headset had a built in design based on international 10-20 system (Fp1, Fp2, F7, F3, Fz, F4, F8, T3, C3, Cz, C4, T4, T5, P3, Pz, P4, T6, O1, and O2) positioned according to international 10–20 system .”
Comment 11: Figure 4: Topomaps have been created using which software?
Response 11: We thank you for reviewing our paper and providing us with appropriate comments to improve our paper.
We used a built in sLoreta function in iSyncBrain software (iMediSync, Inc., South Korea). We made sure to mention this in the paper too. This information has been added as follows:
(Currently line 201-203): “The EEG band power was shown in topographic map (Topo map) plotted with built in sLoreta function in iSyncBrain software (iMediSync, Inc., South Korea).”
Comment 12: L218 – L252 does not fit into the conclusion sections, it belongs more to Introduction Literature review. Probably for the literature review: The benefits of emotion regulation interventions in virtual reality for the improvement of wellbeing in adults and older adults: a systematic review (https://www.mdpi.com/2077-0383/9/2/500)
Response 12: Thank-you for your insightful suggestion. We found your comment extremely helpful. The referred the paper was beneficial to get more knowledge regarding Virtual Reality. We agree with your comment and hence we have modified the conclusion accordingly. We added more sentences explaining our study results to provide better discussion. The line 218-161 after revision is currently changed to line 296-338. The revised lines are underlined as follows:
“Our study examined the effect of VR game intervention on cognitive and brain function in older adults with MCI. Our finding shows that VR game intervention is an effective way to improve cognitive and frontal brain function in MCI patients. The improvements in 8ft up and go test and gait speed were also observed in intervention group.
Over the past decade, technology-based cognitive interventions have gained immense interest worldwide. Our study results shows small-to-medium positive effects for VR intervention on key outcome variables such as physical function and cognition. The VR intervention effect in the cognition category was small to medium, which is consistent with the result of a systematic review by Coyle et al. 2015 [31]. Coyle and colleagues showed that VR intervention moderately improved the cognitive function of participants with cognitive impairment. A recent VR study has also reported improvements in executive functions [32] after VR intervention similar to our study results. However our study showed positive improvements in executive performance in TMT B significantly but not in TMT A. TMT A measures psychomotor speed, visual scanning while TMT B reflects working memory [33-35]. The contents of our VR training because the games such as love home, Juice making and fireworks which required subjects to memorize objects, recipes and numbers respectively highly involving use of working memory along with other cognitive functions such as attention and processing speed. This could explain the significant improvement in TMT B score. In this study we measured brain wave activities during the resting state of mind. The theta power was observed to be decreased in the VR intervention in follow up. Increased theta, in elderly, is associated with risk of developing cognitive impairment [28, 29]. Our study also showed significant decrease in TBR in temporal and parietal region of brain. The higher TBR was observed to be related to mind wandering which is considered to be associated with reduced attention [35]. However, detailed studies over a longer period of time on each EEG rhythms are required to draw strong its relationship with VR training.
Studies have also shown that VR intervention is effective to improve physical function and walking speed of community-dwelling patients who had a stroke [36-38]. Our study results further expands this data as significant improvements in physical function especially gait speed and 8ft Up and go test in MCI patients were observed after 8 weeks of VR training. Several studies have found physical functions such as hand grip strength [39-41] and gait speed [42] to be associated with cognitive decline in older adults. Furthermore lower TUG scores and slower gait speed have been reported to be associated with lower performing executive function [43]. Therefore the improvement in physical function in our study could be indicative of improved cognitive function.
Formal issues
Comment 1: Title lower/upper letter mix: should be completely in TitleCase:
The Effect of Virtual Reality (VR)-based Intervention Program and cognition in older adults with Mild Cognitive impairment (MCI): A randomized control trial -> The Effect of Virtual Reality (VR)-based Intervention Program and Cognition in Older Adults with Mild Cognitive Impairment (MCI): A Randomized Control Trial
Response 1: Thank you for your excellent observation. The other reviewer also had suggestions for us regarding the title.
After considering both suggestions, the title now reads as “The Effect of Virtual Reality-based Intervention Program on Cognition in Older Adults with Mild Cognitive Impairment: A Randomized Control Trial”
Comment 2: Title: The effect of VR on Cognition … instead of The effect of VR and cognition?
Response 2: The correction has been made.
Comment 3: Author: Hyuntae park -> Hyuntae Park ?
Response 3: The correction has been made.
Comment 4: L74: “The intervention group performed of VR- based cognitive training for eight weeks.” -> “The intervention group performed an VR- based cognitive training for eight weeks.”
Response 4: Thank you for the suggestion. There other change has also been made to this sentence following suggestion from other reviewer.
The sentence now reads as (Currently line 134-135) “The intervention group performed total 24 sessions of a VR- based cognitive training for eight weeks.”
Comment 5: L82: “The container is until juice is made and poured into the cup.” What does this mean? I do not understand.
Response 5: Thank you for your honest comments. We deeply regret our choice of words.
We have rephrased the sentence as follows (Currently line 148-149): “The container was vigorously shaken until juice was made and then poured into the cup.”
Comment 6: L111: “TMT and DSST were both timed test where shorter time taken to complete the test indicated better cognitive function” Please rephrase.
Response 6: The sentence has been rephrased and now reads as (Currently line186-187) “TMT and SDST were both timed and shorter time taken to complete the test indicated better cognitive function.”
Comment 7: L172 „3.2 EEG\“ -> „3.2 EEG“
Response 7: The correction n has been made.
Comment 8: Please pay attention to the spelling (the following mistakes stem only from L214 to L231, there are many other easy-to-avoid errors)
Response 8: Thank you for screening spelling mistakes in our paper. This comment was extremely helpful in detecting unnoticed spelling mistakes.
We have carefully revised the spellings throughout the manuscript.
Comment 9: L214: “MCI patient” -> “MCI patients”, further: “In previous study“ -> “In a previous study” or “In previous studies”?
Response 9: The correction has been made accordingly.
Comment 10: L224: “might be due the to contents” -> “might be due to the contents”
Response 10: The correction has been made.
Comment 11: L225 “fireworks, had high demand for the use of cognititve funcitons such asworking memory and attention” -> “fireworks, had high demand for the use of cognitive functions such as working memory and attention”
Response 11: Thank you for the comment. The other reviewer also had some suggestions regarding this content.
We have revised the sentence. This sentence after modification reads as (Currently line 315-318) “The contents of our VR training consisted of the games such as love home, Juice making and fireworks which required subjects to memorize objects, recipes and numbers respectively highly involving use of working memory along with other cognitive functions such as attention and processing speed.”
Comment 12: L226 “and processing speed.A recent VR study has aslo reported improvements in executive fucntions [14]” -> “and processing speed. A recent VR study has also reported improvements in executive functions [14]”
Response 12: The correction has been made accordingly.
Comment 13: L226 “In this study we measued brian wave activites during resting rate of mind.” -> “In this study we measured brain wave activities during the resting state of mind.”
Response 13: The correction has been made accordingly.
Comment 14: L231: “Our finding show“ -> “Our findings show” or “Our finding shows”?
Response 14: The correction has been made.
Comment 15: Please mind double spaces (e.g. in L240)
Response 15: The paper has been revised following the reviewers suggestion regarding double spaces.
Comment 16: Careful proofreading, preferable by a native speaker, is required.
Response 16: We agree with this and have proofread the manuscript by our native speaker colleague

Reviewer 2 Report
General review
Thank you very much for the opportunity to review the article "The Effect of Virtual Reality (VR) -based Intervention Program and cognition in older adults with Mild Cognitive impairment (MCI): A randomized control trial " As for the novelty of the article, it stands at a very high level. At the moment, there is a boom in topics related to VR in scientific research and this trend will intensify. That is why I would like to congratulate you on taking up the topic and carrying out valuable research. However, the article itself is written messily. There are many errors of all kinds, e.g. from translation errors, through inconsistencies in describing the number of participants, to the vague presentation of research results. Certainly, the article should undergo professional proof-reading, because there should be no 'imrove' or 'traning' errors in print. Certainly, the strength of the article is EEG use as one of the outcome measures. However, my ability to interpret EEG is limited. Another reviewer who is more familiar with these analyzes may have additional comments, however. Once again, I congratulate the authors of the article, however, it still requires a lot of work before publication.
Detailed review:
- Title: Using acronyms in the title is not good practice. I suggest removing VR and MCI from the title and suggest paying attention to uppercase and lowercase letters
- The last author's last name is lowercase
- Line 17 - please expand MCI to Mild Cognitive Impairment (MCI)
- Line 29 - please do not use abbreviations in keywords
Introduction:
- The introduction could be better written. The authors devoted very little space to outline why immersive VR, from the theoretical side, can be an effective medium in cognitive training. In line 41-42, the authors begin to write about it, but they do not elaborate. It is also worth remembering that the authors write about immersive VR, not non- immersive VR, because immersive VR gives the feeling of ' being there ' or ' presence '.
- Line 33-34 - the first sentence is missing a logical string
- Line 34-35 - the second sentence is also missing a logical string
- Lime 50-54 - this paragraph is very complicated. This is a long sentence with language errors and mental In addition, the authors once again put the abbreviation of MCI. However, in the Line 50 is a very important statement: ' the potential to be powerful tools in delaying onset of degenerative brain and mental diseases'. I propose to develop the ground for undertaking this research. What's more, so far VR is not used as a prevention tool, but rather as a way of training. There is still a long way to go to preventive examinations.
- Line 55 – I suggest replacing the word 'impact' with 'effect'.
Methods:
- Such a study is required to be registered in one of the recognized repositories of clinical trials, even retrospectively. Clinicaltrial.gov allows you to register completed studies.
- Line 61 - Please expand the method for determining MCI. This description of the MCI diagnosis is not sufficient, because if someone wanted to repeat your examination, should know the exact criteria.
- Line 64 - the authors have identified wrote, "our study included a total of 75 subjects." However, in fig. 1. It is written that there were included 68.
- Line 65 - please specify the randomization method.
- Line 67 - could you write something more about the International Review Board? And did you mean the Institutional Review Board?
- Figure 1 - please check the number of people randomized. Please also move the number of people who resigned (32 or 25) one level below, i.e. under recruited section, because they dropped out after this stage. Please specify the reasons for dropouts during your intervention.
- In figure 1 post-test is defined as 'follow up after 8 weeks'. Follow-up provides valuable information about the long-term, however, the description is clear, that the authors did only pre/post design.
- Line 77 - please write the total number of VR sessions held by the participant.
- If you put citation to MMSE and NCGG-FAT, also put citation to TMT-A/B and SDST
- Line 107 - Unify the term SDST/This is described in various ways throughout the article. Even in this paragraph, there is a reference to DSST at the end and SDST at the beginning.
- Intervention - VR intervention is well described. However, there is not a word about what the control group was doing. This is a major methodological flaw.
- Line 132 - TUG should be performed at 3 meters distance.
- Line 137 - the authors wrote that Body impedance analysis was performed, but nowhere in the text did they refer to the BIA results.
Results :
- The results are quite briefly described. Table 2 describes the results of this study, however, the method of its construction prevents the reader from interpreting the data independently, because there is insufficient information.
- Table 1 - please specify decimals and stick to the entire article. If it's 26, give it 26.0 or 26.00
- Table 1 - please provide p-Value for all parameter differences.
- Table 1 - please correct the units - age and education have the same unit (years) and are marked differently; No. of medication intake (n); mg/Hg uppercase; put space after Weight.
- Line 167 - again another way of referring to what I understand
- Table 2 - is the most important element of this publication. I propose to change the entire table including effect size, confidence interaval, and p-Value. This arrangement will be much more readable when it comes to the effectiveness of your intervention. Please do not use follow-up, only post-test and refine
- Line 166-167 - if something was not statistically significant, I suggest not to write about it in the results.
- If I count correctly, figure 3 has disappeared somewhere or the numbers are incorrectly numbered.
Discussion:
- It is not easy for me to refer to the discussion because Table 2 does not provide sufficient information on the results of the research. Certainly, the discussion lacks an explanation of the mechanisms of the results obtained. Why was TMT-B significantly reduced and TMT-A not? In general, why TMT-B - maybe it was influenced by Fireworks? This is lacking in the discussion. In places, it is written like an introduction.
- Line 234 - imrove
- Line 240 - traning
- Line 255 - lasted mintues long
- Line 256 - cognittive
- Line 258 - in the figure 1, you wrote that there was 1 dropout in both groups, and here you write that 1:6.
- Line 260-262 - limitation section should be expanded. 8-week training is not a study limitation. But no follow-up after 4-8-12 weeks after the survey certainly is. I do not know what intervention the control group had. If controls didn't have one, that is a big limitation of the research.
- At the end of the discussion, I suggest clearly defining the conclusion of your results.
Author Response
Reviewer #2
Comment 1: Title: Using acronyms in the title is not good practice. I suggest removing VR and MCI from the title and suggest paying attention to uppercase and lowercase letters
Response 1: Thank you for this suggestion. We have removed acronyms as suggested by the reviewer.
Comment 2: The last author's last name is lowercase
Response 2: We are thankful to you for reviewing our paper thoroughly. We corrected authors’ last name.
Comment 3: Line 17 - please expand MCI to Mild Cognitive Impairment (MCI)
Response 3: We have expanded MCI to Mild Cognitive Impairment (MCI) in revised manuscript.
Comment 4: Line 29 - please do not use abbreviations in keywords
Response 3: We have removed abbreviations in keywords. We will be mindful of this suggestion in the future too.
Introduction:
Comment 1: The introduction could be better written. The authors devoted very little space to outline why immersive VR, from the theoretical side, can be an effective medium in cognitive training. In line 41-42, the authors begin to write about it, but they do not elaborate. It is also worth remembering that the authors write about immersive VR, not non- immersive VR, because immersive VR gives the feeling of ' being there ' or ' presence '.
Response 1: Thank you for devoting your time to review our paper and provide us with meaningful comment. We appreciate this insightful feedback and agree with it. The introduction does seem lacking in elaboration regarding to why immersive VR can be an effective cognitive training. We have taken your advice while modifying the introduction section. We have made modifications through the introduction section as follows (Currently line 53-76):
“Virtual reality (VR) is a computer stimulated environment, closely resembling real-life situations and scenarios, which provides user with sensation of physically “being there” [13,14]. VR can be divided into non- immersion, semi-immersion, and full-immersion based on levels of immersion. Immersive VR provides enhanced ecological validity and possibility to personalize the activity and environment to the need of individual making VR-based training more engaging [15].The higher level of immersion corresponds to more realistic VR environment to the user [16]. The immersive VR have been used mostly for neurophysiological assessment, cognitive rehabilitation and effect of VR on different cognitive domains such as executive function, attention, memory and spatial orientation is being investigated [17].
Immersive VR based cognitive training has recently attracted attention in research filed of MCI and dementia [18]. A recent Meta analytical study have also reported positively effect of semi-immersive VR on cognition and physical function in individuals with MCI and dementia [19]. Fully immersive VR based cognitive intervention study on older adults with and without dementia reported significantly higher cognitive progress in those without dementia compared to those with dementia [20]. A feasibility study using fully immersive VR cognitive training in MCI and dementia patients showed high feeling of security, satisfaction and reduced anxiety, fatigue, and low discomfort compared to pen-paper training [15]. This advantage of VR over pen-paper cognitive training can be beneficial to ensure adherence to cognitive training and when performed with caution, it has potential to be an efficient intervention for dementia patient [15,21].”
Comment 2: Line 33-34 - the first sentence is missing a logical string
Line 34-35 - the second sentence is also missing a logical string
Response 2: Thank you for the constructive comment. The lines after modification have become line 32-34. The sentences now reads as follows:
(Currently line 35-38): “Dementia is a syndrome characterized by global cognitive impairment [1] and an estimated 50 million people worldwide have dementia [2]. It is mostly prevalent in individuals aged >65 years and is considered as the greatest health challenge in 21 century [3].”
Comment 3: Line 50-54 - this paragraph is very complicated. This is a long sentence with language errors and mental In addition, the authors once again put the abbreviation of MCI. However, in the Line 50 is a very important statement:' the potential to be powerful tools in delaying onset of degenerative brain and mental diseases'. I propose to develop the ground for undertaking this research. What's more, so far VR is not used as a prevention tool, but rather as a way of training. There is still a long way to go to preventive examinations.
Response 3: Thank you for reviewing our paper and providing us with immensely thoughtful comments. In order to simplify the complicated paragraph we have rephrased the previously line 50-54 as follows:
Currently line (80-89) “Nevertheless, immersive VR is increasingly used in health-related fields and interventions and have the potential to be powerful tools in delaying onset of degenerative brain and mental diseases. However the evidences regarding effectiveness of immersive VR in MCI and dementia is small [22], especially the fully immersive type of VR and the advantages of VR with full immersion and interaction on prevention of dementia haven’t been yet explored to full potential.”
We agree with your comment that VR has been used mostly as way of training and indeed more studies are required regarding the use of VR as preventive intervention.
Comment 4: Line 55 – I suggest replacing the word 'impact' with 'effect'.
Response 4: Thank you for your comment. We thank you for highlighting the mistakes in our paper.
The correction has been made as suggested.
Methods:
Comment 1: Such a study is required to be registered in one of the recognized repositories of clinical trials, even retrospectively. Clinicaltrial.gov allows you to register completed studies.
Response 1: Thank you for reviewing our paper and pointing this out.
We have registered our study in the “University Hospital Medical Information Network (UMIN)” Clinical Trials Registry (Registration No. UMIN000040107).”
We have added this information in the method section of manuscript as follows (line 125-126): “This study is registered in the University Hospital Medical Information 124 Network (UMIN) Clinical Trials Registry (Registration No. UMIN000040107).”
Comment 2: Line 61 - Please expand the method for determining MCI. This description of the MCI diagnosis is not sufficient, because if someone wanted to repeat your examination, should know the exact criteria.
Response 2: We deeply appreciate in depth review of our paper and your constructive comment. We agree with your comment on lack of description of MCI diagnosis. We have found your comment to be tremendously helpful to explain our study methods more clearly.
We have expanded the methods for determining MCI in method section as follows:
(Currently line 111-114): “individuals diagnosed with MCI based on medical evaluations consisting of neurological examinations and detailed neuropsychological assessments conducted by a dementia specialist.”
Comment 3: Line 64 - the authors have identified wrote, "our study included a total of 75 subjects." However, in fig. 1. It is written that there were included 68.
Response 3: Thank you for pointing this out. It was an error. We have corrected the error throughout the manuscript.
Comment 4: Line 65 - please specify the randomization method.
Response 4: We have specified the randomization method in the method section.
Comment 5: Line 67 - could you write something more about the International Review Board? And did you mean the Institutional Review Board?
Response 5: We have expanded The IRB information.
(Currently line 123-125): “The study procedure was approved by Dong-A university Institutional Review Board (IRB No. 2-1040709-AB-N-01-201909-HR-036-04) and all participants provided signed informed consent in the beginning of the study.)”
Comment 6: Figure 1 - please check the number of people randomized. Please also move the number of people who resigned (32 or 25) one level below, i.e. under recruited section, because they dropped out after this stage. Please specify the reasons for dropouts during your intervention.
Response 6: Thank you for your comment. We received comments from other reviewer as well regarding the unmatched subject number and missing dropout reasons. We have edited the flowchart with correct number of subjects. We also replaced the word dropout with “didn’t join due to personal reasons” as they were not part of randomization. The reason for not participating has been added in the figure 1 as follows:
- 15 subjects didn’t join due to not accessible location
- 12 subjects didn’t join due to lack of time
- 4 subjects didn’t join due to unexpected health problem (self or family)
- 1 subject moved out of town
Comment 7: In figure 1 post-test is defined as 'follow up after 8 weeks'. Follow-up provides valuable information about the long-term, however, the description is clear, that the authors did only pre/post design.
Response 7: Thank you for your assessment. We deeply appreciate the time and effort you put on to reviewing our paper.
Comment 8: Line 77 - please write the total number of VR sessions held by the participant.
Response 8: We have added total number of VR sessions in the method section as follows:
(Currently line 134-135): “The intervention group performed total 24 sessions of VR- based cognitive training for eight weeks.”
Comment 9: If you put citation to MMSE and NCGG-FAT, also put citation to TMT-A/B and SDST
Response 9: Thank you for the comment. We have added the citation.
Comment 10: Line 107 - Unify the term SDST/This is described in various ways throughout the article. Even in this paragraph, there is a reference to DSST at the end and SDST at the beginning.
Response 10: Thank you for your comment. The correction has been made accordingly throughout the manuscript.
Comment 11: Intervention - VR intervention is well described. However, there is not a word about what the control group was doing. This is a major methodological flaw.
Response 11: We have added the activities performed by the control group in the method section. We apologize for not mentioning the activities of control group.
Line 138-142: “The control group participated in educational program on general health care once a week for 8 weeks. Each class was 30 minutes long. The program was led by health professionals employed at the health care center and subjects were given various information such as, nutrition regarding proper diet and foods, exercise tips to maintain physical function like balance, gait speed, muscle strength, and mental health to prevent dementia.”
Comment 12: Line 132 - TUG should be performed at 3 meters distance.
Response 12: We agree with your comment. We performed 8ft up and go (2.44 m) as a part of Short physical performance battery (SPPB) test in our study. We deeply regret for wrongly naming 8ft up and go test as TUG. We have made correction throughout the manuscript.
Comment 13: Line 137 - the authors wrote that Body impedance analysis was performed, but nowhere in the text did they refer to the BIA results.
Response 13: Thank you for the comment. We removed Body impedance analysis from the methods as we didn’t use the BIA result in our study.
Results:
Comment 1: The results are quite briefly described. Table 2 describes the results of this study, however, the method of its construction prevents the reader from interpreting the data independently, because there is insufficient information.
Response 1: We agree with this, therefore we incorporated additional information in table 2.
Comment 2: Table 1 - please specify decimals and stick to the entire article. If it's 26, give it 26.0 or 26.00
Response 2: The correction has been made as suggested by the reviewer.
Comment 3: Table 1 - please provide p-Value for all parameter differences.
Response 3: Thank you for your thoughtful comment.
We did not observe any significant differences between intervention and control groups in all variables, hence p-values were omitted in table 1.
Comment 4: Table 1 - please correct the units - age and education have the same unit (years) and are marked differently; No. of medication intake (n); mg/Hg uppercase; put space after Weight.
Response 4: Thank you for catching the errors. The correction has been made with proper spacing and units in table 1.
Comment 5: Line 167 - again another way of referring to what I understand
Response 5: We have removed words mentioning insignificant result as advised by the reviewer.
Comment 6: Table 2 - is the most important element of this publication. I propose to change the entire table including effect size, confidence interval, and p-Value. This arrangement will be much more readable when it comes to the effectiveness of your intervention. Please do not use follow-up, only post-test and refine
Response 6: Thank you for pointing out this important point. We have taken your advice and modified table 2. In order to avoid confusion the results without significance are presented as “not significant (n.s.)”
Comment 7: Line 166-167 - if something was not statistically significant, I suggest not to write about it in the results.
Response 7: We are thankful for your invaluable suggestion. We have removed words mentioning insignificant result. We will remember this advice in future too.
Comment 8: If I count correctly, figure 3 has disappeared somewhere or the numbers are incorrectly numbered.
Response 8: Thank you for highlighting the error in our paper. We have made correction in the revised manuscript.
Discussion:
Comment 1: It is not easy for me to refer to the discussion because Table 2 does not provide sufficient information on the results of the research. Certainly, the discussion lacks an explanation of the mechanisms of the results obtained. Why was TMT-B significantly reduced and TMT-A not? In general, why TMT-B - maybe it was influenced by Fireworks? This is lacking in the discussion. In places, it is written like an introduction.
Response 1: Thank you for reviewing our paper and providing us valuable feedback.
We have modified the discussion and provided additional information regarding the reviewer’s question “Why was TMT-B significantly reduced and TMT-A not? In general, why TMT-B - maybe it was influenced by Fireworks?”
Line 311 -319: “However our study showed positive improvements in executive performance in TMT B significantly but not in TMT A. TMT A measures psychomotor speed, visual scanning while TMT B reflects working memory [33-35]. The contents of our VR training because the games such as love home, Juice making and fireworks which required subjects to memorize objects, recipes and numbers respectively highly involving use of working memory along with other cognitive functions such as attention and processing speed. This could explain the significant improvement in TMT B score.”
Comment 2: Line 234 - imrove
Line 240 - traning
Line 255 - lasted mintues long
Line 256 – cognittive
Response 2: Thank you for the thorough review of your paper. The correction has been made accordingly.
Comment 3: Line 258 - in the figure 1, you wrote that there was 1 dropout in both groups, and here you write that 1:6.
Response 3: Thank you for mentioning this error. We have made corrections in the revised manuscript.
(Currently line 363-364): “The subjects in VR intervention group strongly adhered to the training and the dropout number was low (n=1).”
Comment 4: Line 260-262 - limitation section should be expanded. 8-week training is not a study limitation. But no follow-up after 4-8-12 weeks after the survey certainly is. I do not know what intervention the control group had. If controls didn't have one, that is a big limitation of the research.
Response 4: We greatly appreciate your thoughtful comment. We have modified the limitations.
(Currently line 366-382): “One of the limitation of our study was that there was no follow up in middle of intervention period or some period after the study was over due to which the short and long term effect of VR intervention on cognition and physical function were not observed. Secondly VR intervention program group had more sessions (n=24) compared to control group (n=8) and hence intervention group had more interaction with each other and their supervising health professionals. The increased social interaction has been associated with preventing cognitive decline [53, 54]. This could have somewhat contributed to positive result in intervention group. The participants of our study were predominantly women (n=52) compared to men (n=16) and future studies regarding gender difference on effect of VR intervention is necessary.
Comment 5: At the end of the discussion, I suggest clearly defining the conclusion of your results.
Response 5: We have added conclusion in the end of paper following reviewer’s advice.
Line 387-390: In summary, our results show that VR- based cognitive training improved cognitive function and physical function in elderly subjects with Mild Cognitive impairment (MCI) and suggests that promoting VR intervention for cognitive training for may prevent or delay progression of cognitive impairment. Nevertheless, further work is needed in this area to confirm the long-term effectiveness and feasibility.

Round 2
Reviewer 1 Report
Thank you very much for doing a good job of accommodating the comments of both reviewers.
However, there are still some comments:
- Non-Treatment of control group: Ok, the control group received an entertainment program, too. However, the effects of VR are still compared with no treatment. So, the statement made is “VR may have a positive effect”. As now the study cannot be changed afterwards, it would be fair to add a sentence to the discussion that it remains to further work to compare the effect sizes achievable with other treatments.
- “the random allocation was centralized and internet-based” -> “internet-based”? Why do need the internet? What is the function of the internet for allocation?
- Proofreading is still recommended. Please check some sample suggestions in the documentation attached.

Author Response
We sincerely appreciate all valuable comments and suggestions, which helped us to improve the quality of the article. Our responses to the Reviewers’ comment are described below in a point-to-point manner. Appropriated changes, suggested by the Reviewers, has been introduced to the manuscript (marked track change in the DOC document).
Comment 1: Non-Treatment of control group: Ok, the control group received an entertainment program, too. However, the effects of VR are still compared with no treatment. So, the statement made is “VR may have a positive effect”. As now the study cannot be changed afterwards, it would be fair to add a sentence to the discussion that it remains to further work to compare the effect sizes achievable with other treatments.
Response 1: Thank you for the suggestion.
We agree with the reviewer and have incorporated the suggestion in the discussion section as follows (Line 377-380):
“Our study yielded a positive effect of fully immersive VR training on cognitive as well as physical function in older adults with MCI. However the VR intervention group was compared with the control group who only received an educational program. Hence future studies should consider comparing the effect of VR intervention with other cognitive treatment in MCI patients.”
Comment 2: “the random allocation was centralized and internet-based” -> “internet-based”? Why do need the internet? What is the function of the internet for allocation?
Response 2: Thank you for the comment. We explained that we performed the random allocation using a fixed computer program via server. Therefore, we modified the content as follows.
To avoid selection bias, the random allocation was assigned via a computer-generated, fixed block randomization procedure to either intervention or control group, generating blocks stratified by age, gender, education level, and participating center.
Comment 3: Proofreading is still recommended. Please check some sample suggestions in the documentation attached.
Response 3: Thank you for the comment. The sample file was extremely helpful for proofreading the manuscript. Furthermore, the grammar and spelling check has also been performed with “Grammarly” tools and software.
Reviewer 2 Report
General review
The authors referred to most of the amendments. However, many issues remain to be resolved. Certainly, the article must undergo professional proof-reading, without that I don't see the chance of publication. The number of typos is too large. Also, many prepositions are missing. First.
Secondly, there was an ethical doubt. Author Hansol Kim (Dong-A University) has been replaced by KyungWon Park (Dong-A University Hospital) in the revised version of the manuscript. The authors did not explain this change. It appeared without justification. The removed author was not listed in the acknowledgments section. Editors will handle this issue.
Detailed review
- In page 1, the JCM logo disappeared
- Line 7-8 should not ‘univesity’ be capitalized?
Abstract
- Line 23 change to plural: ‘comprised four types’
- Line 20 correct SDGT
Introduction
- Line 35-36 ‘…resulting in gravitation toward non-pharmacological approaches to alter the progressive course of disease [4].’ Cited literature: D’Cunha et al ‘A mini-review of virtual reality-based interventions to promote well-being for people living with dementia and mild cognitive impairment’ – this literature does not support the above statement. D’Cunha et al cited two large systematic reviews.
Minor issues
- Line 43 – change ‘trail’ -> ‘trial’
- Line 38 – ‘some MCI patients’
- Line 47 – stimulated? Rather simulated
- Line 51 – put space before ‘The’
- Line 57 – Replace ‘A recent Meta analytical study have…’ to ‘A recent meta-analysis has…’ or similar
- Line 57 – change ‘positively’ to ‘positive’
- Line 65 – ‘dementia patients’
- Line 67 – immersive VR…has potential’
- Line 68 – noun ‘evidence’ seems to be not countable
- Line 68 – I don’t believe ‘small’ is a good word in this context
Methods
Once again: please expand the method for determining MCI. An important issue of scientific research is the ability to replicate it. Specifying the inclusion criteria, in this case MCI, is extremely important for someone who would like to repeat your research. I mean, for example, an MMSE score <24/27 etc. Please, specify exactly how the psychologist chose the participants. Was it the result of MMSE or TMT or something else.
- I suggest a slightly clearer demarcation of interventions in the experimental group and the control group.
- Line 81 – criteria word is plural from criterion
- Line 91- ‘consent at the’
- Line 110 – ‘controllers’
- Line 111 – ‘requires’
- Line 135 – correct to ‘Mini-Mental State Examination-Dementia Screening test’
I stopped at his point correcting spelling errors.
Results
- Once again, correct decimal places in tables (mean, SD).
- Line 194 – SDGT once again
- Line 200 – SDGT once again
- Table 2 - SDGT once again
- Table 2 – to what refers *?
- 210-211 – SDGT and TUG
Conclusion
I suggest you soften the conclusion gently. First, the control group had significantly less intervention. Secondly, MMSE appeared to be the main outcome measure in which no statistically significant change was obtained. I also propose to include a sentence regarding EEG in the conclusion.
Authors contributions
- Remove generic text from this section, correct authors contribution
Author Response
We sincerely appreciate all valuable comments and suggestions, which helped us to improve the quality of the article. Our responses to the Reviewers’ comment are described below in a point-to-point manner. Appropriated changes, suggested by the Reviewers, has been introduced to the manuscript (marked track change in the DOC document).
Comment 1: The authors referred to most of the amendments. However, many issues remain to be resolved. Certainly, the article must undergo professional proof-reading, without that I don't see the chance of publication. The number of typos is too large. Also, many prepositions are missing. First.
Response 1: Thank you for the comment.
The spelling, grammar, and punctuation were proofread using Grammarly software. The manuscript was further proofread to check for other possible errors.
Comment 2: Secondly, there was an ethical doubt. Author Hansol Kim (Dong-A University) has been replaced by KyungWon Park (Dong-A University Hospital) in the revised version of the manuscript. The authors did not explain this change. It appeared without justification. The removed author was not listed in the acknowledgments section. Editors will handle this issue.
Response 2: Thank you for your concerns. We deeply apologize for not addressing the change in authorship. The change in authorship was agreed upon by both authors (Dr. KyungWon Park and Ms. Hansol Kim). This issue has been addressed in the email with the assistant editor.
We have also expressed our thanks to Ms. Hansol Kim in the acknowledgment section in this revised manuscript.
Detailed review
Comment 1: In page 1, the JCM logo disappeared
Response 1: The logo was erased by mistake. The JCM logo has been added to the first page of the manuscript.
Comment 2: Line 7-8 should not ‘university’ be capitalized?
Response 2: The correction has been made.
Abstract
Comment 1: Line 23 change to plural: ‘comprised four types’
Response 1: The correction has been made. We used the Grammarly software to correct grammar errors and spelling.
Comment 2: Line 20 correct SDGT
Response 2: The correction has been made.
Introduction
Comment 1: Line 35-36 ‘…resulting in gravitation toward non-pharmacological approaches to alter the progressive course of disease [4].’ Cited literature: D’Cunha et al ‘A mini-review of virtual reality-based interventions to promote well-being for people living with dementia and mild cognitive impairment’ – this literature does not support the above statement. D’Cunha et al cited two large systematic reviews.
Response 2: Thank you for pointing this out. We have cited the sentence with correct reference in the revised version.
Minor issues
Comment 1: Line 43 – change ‘trail’ -> ‘trial’
Response 1: The correction has been made.
Comment 2: Line 38 – ‘some MCI patients’
Response 2: The correction has been made.
Comment 3: Line 47 – stimulated? Rather simulated
Response 3: Thank you for the appropriate suggestion. The correction has been made.
Comment 4: Line 51 – put space before ‘The’
Response 4: The correction has been made.
Comment 5: Line 57 – Replace ‘A recent Meta analytical study have…’ to ‘A recent meta-analysis has…’ or similar
Line 57 – change ‘positively’ to ‘positive’
Response 5: Thank you for the comment. The correct has been made and the sentence now reads as (Line 66-67)
“A recent meta-analytic study has also reported the positive effect of semi-immersive VR on cognition and physical function in individuals with MCI and dementia”.
Comment 6: Line 65 – ‘dementia patients’
Response 6: The correction has been made.
Comment 7: Line 67 – immersive VR…has potential’
Response 8: The correction has been made.
Comment 8: Line 68 – noun ‘evidence’ seems to be not countable
Response 8: Thank you for the comment. We have corrected the word evidences to evidence. The sentence now reads as: “However the evidence regarding the effectiveness of immersive VR in MCI and dementia is limited [22], especially the fully immersive type of VR.”
Comment 9: Line 68 – I don’t believe ‘small’ is a good word in this context
Response 9: Thank you for the appropriate suggestion. We replaced the word “small” with “limited” to suit the context.
Methods
Comment 1: Once again: please expand the method for determining MCI. An important issue of scientific research is the ability to replicate it. Specifying the inclusion criteria, in this case MCI, is extremely important for someone who would like to repeat your research. I mean, for example, an MMSE score <24/27 etc. Please, specify exactly how the psychologist chose the participants. Was it the result of MMSE or TMT or something else?
Response 1: Thank you for highlighting the important issue. We have expanded the methods for defining MCI as follows (line 114-116):
“MCI was defined by the Consortium to Establish a Registry for Alzheimer's Disease Assessment Packet (CERAD) with the cut-off score developed by Chandler et al.”
Reference
Chandler, Melanie J., et al. "A total score for the CERAD neuropsychological battery." Neurology 65.1 (2005): 102-106.
Comment 2: I suggest a slightly clearer demarcation of interventions in the experimental group and the control group.
Response 2: Thank you for the comment.
We have made changes for better separation of interventions in the intervention and the control group.
It reads as “The intervention group performed a total of 24 sessions of VR- based cognitive training for eight weeks. Three sessions were held per week and each VR training session lasted for 100 minutes which also included instruction regarding VR training and eye stretching exercises in between VR training as described in Figure 2. On the other hand, the control group participated in an educational program on general health care once a week during the study intervention period (8 sessions). Each session was 30 minutes to 50 minutes. The program was led by health professionals, exercise specialist, physical therapist and nutritionist, and subjects were given information such as, nutrition regarding proper diet and foods, exercise tips to prevent geriatric diseases such as frailty, sarcopenia, and dementia. In addition to VR training, the intervention group was also provided with an educational program following the same protocol as the control group.”
Comment 3: Line 81 – criteria word is plural from criterion
Line 91- ‘consent at the’
Line 110 – ‘controllers’
Line 111 – ‘requires’
Line 135 – correct to ‘Mini-Mental State Examination-Dementia Screening test’
I stopped at his point correcting spelling errors.
Response 3: Thank you for thoroughly reviewing our manuscript. We used Grammarly software for grammar and spelling check.
Results
Comment 1: Once again, correct decimal places in tables (mean, SD).
Response 1: The correction has been made in table 1 and table 2.
Comment 2: Line 194 – SDGT once again
Line 200 – SDGT once again
Table 2 - SDGT once again
Response 2: We have replaced “SDGT” with “SDST”.
Comment 3: Table 2 – to what refers *?
Response 3: The information has been added to table 2. It reads as “* represents a significant difference between the intervention and control group”.
Comment 4: 210-211 – SDGT and TUG
Response 4: We apologize for not making proper corrections in 1st revision. The correction has been made.
Conclusion
Comment 1: I suggest you soften the conclusion gently. First, the control group had significantly less intervention. Secondly, MMSE appeared to be the main outcome measure in which no statistically significant change was obtained. I also propose to include a sentence regarding EEG in the conclusion.
Response 1: Thank you for the comment. We are grateful for your constructive suggestion.
We have modified the conclusion as follows:
“In summary, our results show positive effects of a VR-based cognitive training on cognition in MCI patients. Although the global cognitive function didn't change significantly, we observed significant improvement in executive function, especially working memory, and some physical functions such as gait speed and 8 feet up and go test. Moreover, the EEG test showed a positive change in brain activity related to attention after the intervention period. Nevertheless, further work is needed in this area to confirm the long-term effectiveness and feasibility.”
Authors contributions
Comment 1: Remove generic text from this section, correct authors contribution
Response 1: Thank you for the comment. The generic text has been removed.